# The Relationship among College Students’ Physical Exercise, Self-Efficacy, Emotional Intelligence, and Subjective Well-Being

**DOI:** 10.3390/ijerph191811596

**Published:** 2022-09-15

**Authors:** Kun Wang, Yan Li, Tingran Zhang, Jiong Luo

**Affiliations:** Research Centre for Exercise Detoxification, College of Physical Education, Southwest University, Chongqing 400715, China

**Keywords:** college students, subjective well-being, physical exercise, self-efficacy, emotional intelligence, chain mediation

## Abstract

Purpose: To deeply analyze the influencing factors on college students’ subjective well-being and the path mechanism between each factor. Method: The physical activity rating scale (PARS-3), the emotional intelligence scale (EIS), the self-efficacy scale (GSES), and the subjective well-being scale (SWS) were used for 826 students from two comprehensive universities in southwest China. College students conducted a questionnaire survey, and used SPSS22.0 and AMOS21.0 statistical software to process and analyze the obtained data. Results: (1) Physical exercise was significantly positively correlated with self-efficacy, emotional intelligence, and subjective well-being; self-efficacy was significantly positively correlated with emotional intelligence and subjective well-being; emotional intelligence was significantly positively correlated with subjective well-being; (2) Physical exercise has a direct positive predictive effect on subjective well-being (ES = 0.16); (3) Self-efficacy (ES = 0.057) and emotional intelligence (ES = 0.077) play a part in mediating the role between physical exercise and subjective well-being, respectively. Meanwhile, the chain mediation effect (ES = 0.026) of self-efficacy and emotional intelligence also achieved a significant level, among them, only others’ emotional management and emotional application were involved in the regulation of emotional intelligence. Conclusions: Actively participating in physical exercise could not only directly improve the level of subjective well-being of college students but also indirectly affect their subjective well-being by improving their self-efficacy, as well as their emotional management and emotional application abilities, thereby improving college students’ life satisfaction feelings of happiness, such as degree, positive emotion, and negative emotion.

## 1. Introduction

At present, with the continuous transition from a survival-oriented society to a development-oriented society, building a happy society has become one of the goals of a country’s future economic and social development. However, among college students, the incidence of mental health problems is increasing year by year [1,2], which greatly reduces the happiness experience of this group. The university period is a key stage for the healthy development of an individual’s physical and mental health and the improvement of personality characteristics, and it faces rapid physical and psychological development and gradually matures [3]. Meanwhile, at this stage, individuals are vulnerable to pressures from study, life, and employment, and often have inner conflicts when dealing with emotional, interpersonal, and other issues [4]. It is easy to cause emotional disorders in individuals for a long time, thereby reducing the level of their mental health [5]. Therefore, improving the mental health level of college students and increasing their happiness experience is the key to ensuring their healthy growth and adapting to social development. In the field of sports research, the word “happiness” was frequently mentioned in national sports policy documents. So, can school physical education improve the happiness of college students? The effect, path, and mechanism of the related proposition “Sports promotes the happiness of college students” thus spawned what has undoubtedly become one of the most attractive hot spots in the field of sports science research.

### 1.1. Research on the Relationship between Physical Exercise and Subjective Well-Being

Subjective well-being is an individual’s overall evaluation of his current quality of life according to his standards; it is an important index to evaluate one’s quality of life, and also a comprehensive psychological index to measure the quality of life, which mainly includes three aspects, life satisfaction, positive emotion, and negative emotion [6,7]. Under the guidance of positive psychology, people gradually realized that improving subjective well-being was an important way to promote their health. Meanwhile, with the development of exercise psychology, more and more studies have confirmed that active physical exercise or exercise participation has a significant positive role in promoting individual subjective well-being [8,9,10]. This promotion effect can be roughly divided into the two aspects of internal and external effects. The former emphasizes the pleasure, fluency, and climax generated during physical exercise, allowing individuals to obtain more exercise pleasure, and significantly improve energy and happiness [11,12]. The latter pays attention to the social utility produced in the process of sports participation, including social communication and interaction with others [13,14], and this relationship could bring positive happiness experiences to participating individuals [15,16]. Some scholars have also proposed that active physical exercise could positively predict the subjective well-being of college students [17]. However, since subjective well-being involves life satisfaction and emotional experience, while physical exercise promotes the improvement of individual subjective well-being through internal and external effects, whether it was directly or indirectly accompanied by changes in other psychological or emotional factors, remains to be explored. Based on this, the research hypothesis (H1) is put forward that physical exercise has a positive predictive effect on the subjective well-being of college students.

### 1.2. Correlation Research on Self-Efficacy, Emotional Intelligence, and Subjective Well-Being

Self-efficacy refers to the subjective evaluation of an individual’s ability or grasp of his ability to complete a specific behavior in a specific situation. It was often used to explain the reasons for motivation in a special situation, could predict and explain the corresponding behavior, and it was also the psychological motivation for the continuous self-regulation of the individual [18]. Self-efficacy was considered to be closely related to subjective well-being; some studies have shown that self-efficacy has a significant positive predictive effect on subjective well-being; and the level of self-efficacy has a linear relationship with the individual’s mental health [19,20,21]. Simultaneously, the personality trait theory of subjective well-being holds that people’s happy or unhappy genetic qualities enable people to experience life positively or negatively, that is, lifestyle and emotional experience are highly correlated [22], so, does this also imply that emotional intelligence is closely related to subjective well-being? Emotional intelligence refers to the expression and evaluation of the emotions of oneself and others, the ability to control the emotions of oneself and others, and the ability to use emotions to solve practical problems, that is, a comprehensive ability to accurately perceive, express, and evaluate emotions [23]. Since subjective well-being is a high-level emotion or emotional experience, and emotional intelligence is manifested in the individual’s ability to recognize, regulate, and manipulate emotional information, can emotional intelligence effectively predict subjective well-being? In recent years, studies have shown that an individual’s emotional intelligence was significantly positively correlated with their subjective well-being, and emotional intelligence can positively predict subjective well-being [22,24,25]. Therefore, we speculate that both self-efficacy and emotional intelligence may be important variables that affect subjective well-being. Based on this, the research hypothesis is put forward (H2) that self-efficacy has a positive predictive effect on the subjective well-being of college students and (H3) that emotional intelligence has a positive predictive effect on the subjective well-being of college students.

### 1.3. The Mediating Effect of Self-Efficacy and Emotional Intelligence between Physical Exercise and Subjective Well-Being

Research on exercise psychology has shown that physical exercise behavior and self-efficacy promote each other; active sports participation could positively affect the participants’ sense of self-efficacy; and self-efficacy will gradually increase with the increase in exercise time and exercise level, improve participants’ satisfaction with life, and promote their physical and mental health [26,27,28,29]. Studies have pointed out that there was a significant correlation between physical exercise and the self-efficacy of college students; the higher the degree of physical exercise, the higher the exercise self-efficacy [30]. Meanwhile, the sense of self-efficacy plays a mediating role between physical exercise and the mental health of middle school students; students who participate more in sports have a stronger sense of self-efficacy, often have higher subjective well-being, and the ability to adapt to interpersonal relationships [31,32]. Furthermore, some scholars also pointed out that physical exercise was closely related to emotional intelligence, the individual’s emotional characteristics will be more obvious after physical exercise, and those who consciously insist on long-term exercise generally have a lower chance of occurrence of emotional problems such as anxiety, shock, and social obstacles [29,33]. Participating in sports activities could reduce the tension, anger, and depression of overweight adolescents, and effectively promote their positive health and self-satisfaction [34]. The results suggest that self-efficacy and emotional intelligence may have a positive effect on physical exercise and subjective well-being. In addition, studies have found that there was a correlation between adolescents’ self-efficacy and emotional intelligence, and that self-efficacy could positively predict college students’ emotional intelligence [35,36]. Similarly, some studies have proved that the higher the sense of self-efficacy, the better the performance of college students in emotional intelligence [37]. Based on this, the research hypothesis (H4) is put forward that self-efficacy and emotional intelligence play a chain mediating role between physical exercise and subjective well-being.

In summary, although some studies have revealed the positive effects of physical exercise on the subjective well-being of different groups of people, since subjective well-being was a comprehensive psychological indicator involving many influencing factors, the relationships between different factors have not been validated effectively by the model, and few scholars pay attention to the path of “physical exercise + self-efficacy + emotional intelligence + subjective well-being” of college students. Therefore, this research constructs a chain mediation model between physical exercise, self-efficacy, emotional intelligence, and subjective well-being, which aims to reveal the intrinsic correlations between various variables, and improve the level of college students’ psychological health and well-being.

## 2. Method

### 2.1. Participants

This research focuses on some students from freshmen to senior grades at two comprehensive universities in Chongqing, China, who were selected as the survey objects in this study, and 110 to 130 students in each grade were stratified by random sampling to conduct a questionnaire survey, respectively. We obtained the student IDs of all students from the office in advance, and randomly sampled the student IDs at a ratio of about 1:100 according to the actual number of students in the two schools. The selected students were informed to participate in this research and assigned to the designated classroom to complete the questionnaire. To ensure the authenticity of the questionnaire, we fully explained the content and precautions of the questionnaire to the participants before filling in the questionnaire, and adopted the method of distributing and returning the questionnaire on site. The participants needed to complete the questionnaire in the classroom within 20 min. A total of 1005 questionnaires were distributed, and 948 were recovered, with a recovery rate of 94.33%. After excluding 122 invalid questionnaires such as unclear or uncompleted key information, 826 valid questionnaires were finally obtained, with an effective rate of 87.13%. Among them, the questionnaires included 381 boys (46.1%); 445 girls (53.9%); 259 freshmen (31.4%); 320 sophomores (38.7%); 125 juniors (15.1%); and 122 seniors (14.8%); and their average age was 20.13 ± 1.05 years. This study was approved by the Ethics Committee of Ethics Committee of School of Physical Education, Southwest University (SWU-TY202105) and followed the Declaration of Helsinki, and written informed consent was obtained from all participants.

### 2.2. Questionnaire Design and Reliability and Validity Test

#### 2.2.1. Physical Activity Rating Scale (PARS-3)

The Physical Activity Rating Scale (PARS-3) is designed for categorizing a person’s level of physical activity. The PARS-3 was revised by Liang [38], namely physical exercise intensity, exercise time, and exercise frequency. We used Likert’s 5-point formula for quantification, with a score ranging from 1 to 5 points. The physical exercise score = exercise intensity score × (exercise time score-1) × exercise frequency score, and the score range was 0–100 points, to measure the level of participation in physical exercise. The actual physical activity scores for all participants in this study ranged from 0 to 100. The test–retest reliability of the scale was relatively high, and the correlation coefficient was r = 0.82.

#### 2.2.2. Emotional Intelligence Scale (EIS)

The Emotional Intelligence Scale (EIS) is designed to assess an individual’s ability to express, regulate, and use emotions to solve practical problems. The Chinese version of the EIS, revised by Wang et al., was used [39]. The scale has a total of 33 items, of which 5, 28, and 33 were reverse scoring, which was quantified using the Likert 5-point scale. According to the option “very inconsistent–very consistent”, they were counted as 1–5 points, the score range was 33–165 points, and the higher the score indicated that the emotional intelligence was stronger. The actual emotional intelligence scores of all participants in this study ranged from 42 to 130. After performing factor analysis on the emotional intelligence scale, a total of four common factors were extracted, namely emotional perception (EP); self-emotion management (SEM); emotion management of others (EMO); and emotional application (EA). After direct oblique rotation, the progressive contribution rate of the four common factors was 50.733%. The Cronbach’s α coefficient of EP, SEM, EMO, and EA were 0.88, 0.82, 0.91, and 0.80, respectively. The overall Cronbach’s α coefficient of the scale was 0.91. The confirmatory factor analysis results were: x^2^/df = 1.79, RMSEA = 0.04, AGFI = 0.97, TLI = 0.99, CFI = 0.94, IFI = 0.97, GFI = 0.96, which shows that the scale has good reliability and measurement validity.

#### 2.2.3. General Self-Efficacy Scale (GSES)

The General Self-Efficacy Scale (GSES) is designed for individuals to subjectively assess their ability or degree of assurance that they can perform specific behaviors. The General Self-Efficacy scale was used, revised by Wang et al. [40], which contains a total of 10 questions, which were quantified by a Likert 4-point scale, and were counted as 1 to 4 points according to the option “disagree–strongly agree”, the score range was 10–40 points, and the higher the score, the stronger the sense of self-efficacy. The actual self-efficacy scores for all participants in this study ranged from 10 to 40 points. After direct oblique rotation, one common factor contained a total of nine items, and another one item was eliminated because it contributed too little to the common factor, the progressive contribution rate of each common factor was 54.158%. The overall Cronbach’s α coefficient of the scale was 0.83. The confirmatory factor analysis results were: x^2^/df = 1.96, RMSEA = 0.06, AGFI = 0.95, TLI = 0.98, CFI = 0.97, IFI = 0.98, GFI = 0.98, which shows that the scale has good reliability and measurement validity.

#### 2.2.4. Subjective Well-Being Scale (SWS)

The Subjective Well-Being Scale (SWS) is a scale designed for individuals to evaluate their quality of life as a whole. The SWS contains two subscales of Life Satisfaction (LS) and the Emotion Scale (ES). The Life Satisfaction Scale, compiled by Diener et al. [41], was used to measure the cognitive components of subjective well-being, the scale contains five items, which were quantified by a Likert 7-point scale, which was counted separately according to the option “completely disagree–completely agree”, the score range was 5–35 points, and the higher the score, the higher the life satisfaction (LS). The life satisfaction scores for all participants in this study ranged from 5 to 35. The Cronbach’s α coefficient of this scale was 0.85. The confirmatory factor analysis results were: x^2^/df = 2.12, RMSEA = 0.05, AGFI = 0.91, TLI = 0.97, CFI = 0.96, IFI = 0.98, GFI = 0.95.

The Positive and Negative Affect Scale, revised by Qiu et al. [42], was used to measure the emotional component of subjective well-being. The scale contains a total of 18 items and was quantified using the Likert 5-point scale. The option “nothing at all–very strong” was counted as 1 to 5 points, respectively, and the negative emotion items were scored in reverse, and the score range was 18–90 points. The positive affect and negative affect scores ranged from 20 to 85 for all participants in this study. After performing factor analysis on the scale, a total of two common factors were extracted, namely, positive emotion (PE) and negative emotion (NE). After direct oblique rotation, the progressive contribution rate of the two common factors reached 50.610%. The Cronbach’s α coefficient of PE and NE were 0.82, 0.84, respectively, and the overall Cronbach’s α coefficient was 0.84. Measurement model verification results were: x^2^/df = 1.75, RMSEA = 0.04, AGFI = 0.98, TLI = 0.99, CFI = 0.96, IFI = 0.97, GFI = 0.98. It shows that the scale has good reliability and measurement validity. According to previous studies [3], the total subjective well-being was divided into the sum of Z scores after the reverse scoring of LS, PE, and NE (see Table 1).

### 2.3. Data Analysis

This research used SPSS21.0 and AMOS21.0 to process and analyze the data. Among them, the exploratory factor analysis (EFA), confirmatory factor analysis (CFA), and Cronbach’s Alpha coefficient were used to test the reliability and validity of the scale. The Harman single factor test was used to test the common method deviation of the scale, and the Pearson correlation analysis and linear regression analysis were used to test the relationship between variables. Meanwhile, according to the mediation effect test process proposed by Wen et al. [43], the Test of Joint Significance method was used and AMOS21.0 was used to establish a structural equation modeling to test the mediation effect. The significance level of all indicators was set to *p* < 0.05.

## 3. Research Results

### 3.1. Common Method Deviation Test

In this study, the Harman single factor test method was used to test the common method deviation [44]; the results showed that there were 13 factors with characteristic roots greater than 1, and the variance explained by the first factor was 24.15%, which was less than 40% of the critical standard, indicating that there was no serious common method deviation problem.

### 3.2. Correlation Analysis of Physical Exercise, Self-Efficacy, Emotional Intelligence, and Subjective Well-Being

The Pearson correlation analysis results showed that physical exercise was significantly positively correlated with self-efficacy (*r* = 0.38, *p* < 0.001), and was significantly positively correlated with emotional intelligence (*r* = 0.34, *p* < 0.001), and subjective well-being was significantly positively correlated (*r* = 0.31, *p* < 0.001). Self-efficacy was significantly positively correlated with emotional intelligence (*r* = 0.32, *p* < 0.001), and was significantly positively correlated with subjective well-being (*r* = 0.30, *p* < 0.001). Emotional intelligence and subjective well-being were significantly positively correlated (*r* = 0.31, *p* < 0.001). In addition, the correlation between the main variables and the sub-dimensions all reached a significant level, which provides a good basis for the subsequent test of the mediation effect (see Table 2).

### 3.3. Test of the Mediating Effect of Self-Efficacy and Emotional Intelligence

This study uses the mediation effect test procedure proposed by Wen [43] to conduct a mediation effect test to investigate the relationship between physical exercise, self-efficacy, emotional intelligence, and subjective well-being of college students, and to reveal the mediating role of self-efficacy and emotional intelligence. Firstly, we tested the total effect of physical exercise on subjective well-being, and then tested the fit of the model and the significance of each path coefficient after adding the intermediary variables (self-efficacy, emotional intelligence).

In the total effect model, physical exercise could directly and significantly predict subjective well-being (*β* = 0.32, *p* < 0.001, *SE* = 0.03). After adding the two mediating variables of self-efficacy and emotional intelligence (Figure 1), the path coefficient of physical exercise on subjective well-being decreased from 0.32 to 0.16 (*p* < 0.001, *SE* = 0.02), and all fitting indexes reached the acceptable level, that is x^2^/df = 1.49, RMSEA = 0.02, GFI = 0.99, TLI = 0.98, CFI = 0.97, NFI = 0.96, AGFI = 0.98. The results of the intermediary test showed that physical exercise could significantly positively predict self-efficacy (*β* = 0.38, *p* < 0.001, *SE* = 0.02), emotional intelligence (*β* = 0.32, *p* < 0.001, *SE* = 0.01), and subjective well-being (*β* = 0.16, *p* < 0.001, *SE* = 0.02). Self-efficacy could significantly positively predict emotional intelligence (*β* = 0.29, *p* < 0.001, *SE* = 0.01) and subjective well-being (*β* = 0.15, *p* < 0.001, *SE* = 0.03). Emotional intelligence could significantly positively predict subjective well-being (*β* = 0.24, *p* < 0.001, *SE* = 0.08). In addition, according to the chain intermediary test process proposed by Taylor et al. [45], the test of joint significance (Test of Joint Significance) was used to test the chain intermediary effect of physical exercise on subjective well-being, the results showed that the mediating effect of the path “physical exercise → self-efficacy → subjective well-being” was significant, and the effect value was 0.057. The path of “physical exercise → emotional intelligence → subjective well-being” has a significant mediating effect, the effect value was 0.077. The chain mediation effect produced by the path of “physical exercise → self-efficacy → emotional intelligence → subjective well-being” was significant, and the effect value was 0.026 (see Table 3).

To further reveal the deeper relationship among variables, this study conducted a multiple linear regression analysis on the relationship between each dimension.

First, the path of “physical exercise → self-efficacy → subjective well-being” was tested (Table 4). The regression analysis results of each dimension show that physical exercise has a direct predictive effect on LS (*β* = 0.279, *p* < 0.001), PE (*β* = 0.286, *p* < 0.001), and NE (*β* = 0.294, *p* < 0.001), and physical exercise has a direct predictive effect on self-efficacy (*β* = 0.382, *p* < 0.001). In addition, when physical exercise and self-efficacy were both used as independent variables, physical exercise, and self-efficacy could jointly predict LS (β = 0.210, *p* < 0.001; *β* = 0.180, *p* < 0.001, respectively), and could also jointly predict PE (*β* = 0.208, *p* < 0.001; *β* = 0.205, *p* < 0.001, respectively) and NE (*β* = 0.216, *p* < 0.001; *β* = 0.204, *p* < 0.001, respectively). It suggested that self-efficacy has a partial mediating effect on physical exercise and life satisfaction, positive emotion, and negative emotion, respectively.

Secondly, the path of “physical exercise → emotional intelligence → subjective well-being” was tested (Table 5). The regression analysis results of each dimension showed that physical exercise has a direct predictive effect on LS (*β* = 0.279, *p* < 0.001), PE (*β* = 0.286, *p* < 0.001), and NE (*β* = 0.294, *p* < 0.001), meanwhile, physical exercise has a positive effect on EP (*β* = 0.356, *p* < 0.001), SEM (*β* = 0.185, *p* < 0.001), EMO (*β* = 0.398, *p* < 0.001), and EA (*β* = 0.115, *p* < 0.01). Direct prediction effect: in addition, when physical exercise and self-efficacy (EP, SEM, EMO, EA) were both used as independent variables, besides EP and SEM, the physical exercise, EMO, and EA could jointly predict LS (*β* = 0.198, *p* < 0.001; *β* = 0.165, *p* < 0.01; *β* = 0.084, *p* < 0.05, respectively), and could also predict PE (*β* = 0.201, *p* < 0.001; *β* = 0.168, *p* < 0.01; *β* = 0.105, *p* < 0.01, respectively) and NE (*β* = 0.205, *p* < 0.001; *β* = 0.141, *p* < 0.01; *β* = 0.130, *p* < 0.01, respectively). It suggested that, in the mediating effect of emotional intelligence, only the two dimensions of other people’s emotional management and emotional application play a part in the mediating effect in the path of “physical exercise → emotional intelligence → subjective well-being”.

Finally, “physical exercise → self-efficacy → emotional intelligence → subjective well- being” was tested (Table 6). The regression analysis results of each dimension showed that when physical exercise, self-efficacy, and emotional intelligence (EP, SEM, EMO, EA) were all used as independent variables, in addition to EP and SEM, the physical exercise, self-efficacy, EMO and EA could predict LS together (*β* = 0.162, *p* < 0.001; *β* = 0.136, *p* < 0.001; *β* = 0.136, *p* < 0.01; *β* = 0.090, *p* < 0.05, respectively), and also predict PE together (*β* = 0.159, *p <* 0.001; *β* = 0.161, *p* < 0.001; *β* = 0.135, *p* < 0.01; *β* = 0.113, *p* < 0.01, respectively) and NE (*β* = 0.164, *p* < 0.001; *β* = 0.157, *p* < 0.001; *β* = 0.108, *p* < 0.05; *β* = 0.137, *p* < 0.001, respectively). It showed that among the chain mediation effects of self-efficacy and emotional intelligence, mainly self-efficacy, emotional management of others, and emotional application play a part in the mediating effect, which in turn affects life satisfaction, positive emotions, and negative emotions.

## 4. Discussion

### 4.1. The Direct Impact of Physical Exercise on the Subjective Well-Being of College Students

This study found that physical exercise could positively predict the subjective well-being of college students. This was consistent with previous studies [46,47]. Generally speaking, when an individual has more positive emotions, less negative emotions, and higher life satisfaction, it will be accompanied by an increase in subjective well-being [48]. Studies have shown that sports participation and subjective well-being were positively correlated, and college students who regularly participate in sports activities score higher in life satisfaction and positive emotions [49]. Meanwhile, researchers such as Chen et al. [50] believe that physical exercise has a direct positive predictive effect on the subjective well-being of college students, that is, physical exercise could effectively promote college students’ physical and mental satisfaction, to enhance and improve their sense of pleasure, positive emotions, and subjective evaluation of the quality of life. Among them, campus sports activities have a significant impact on the quality of life of college students [51]. This showed that regular participation in physical exercise could induce positive emotions, improve life satisfaction, and produce positive benefits for participants’ well-being [52,53]. In addition, some scholars pointed out that active participation in sports activities can not only enable people to obtain a “happy and successful psychological experience”, but also achieve individual happiness satisfaction through participation in physical exercise [17,54]. Therefore, physical exercise has a positive predictive effect on the subjective well-being of college students, and college students could improve their subjective well-being experience by actively participating in physical exercise or activities.

### 4.2. The Mediating Effect of Self-Efficacy between Physical Exercise and Subjective Well-Being

This study found, through structural equation modeling, that self-efficacy plays a part in the mediating role in physical exercise and subjective well-being, with a mediation effect ratio of 17.81%, and we proved that self-efficacy has a partial mediating effect on the three dimensions of physical exercise and life satisfaction, positive emotion, and negative emotion. Previous studies have shown that physical exercise plays an important role in enhancing the individual’s sense of self-efficacy, and self-efficacy will change adaptively with the depth of exercise [29,55]. Among college students, physical exercise and self-efficacy were significantly positively correlated; the higher the degree of physical exercise, the higher the college students’ exercise self-efficacy, and there were significant differences in the individual’s self-efficacy for different amounts of exercise [27,30]. Some scholars pointed out that people with high self-efficacy were generally more confident in life and maintained a certain level of subjective well-being, that is, self-efficacy could significantly positively predict subjective well-being [56,57]. In contrast, people with low self-efficacy were often skeptical of their abilities, and tended to choose to relax or give up when facing difficulties, and they had relatively more negative emotional experiences [58], which will reduce their subjective well-being to a certain extent. In addition, some scholars believe that physical exercise, self-efficacy, and subjective well-being were closely related, physical exercise could promote self-efficacy, and high self-efficacy could effectively improve the individual’s life satisfaction, quality of life, and other happiness perception experiences [59,60,61].

### 4.3. The Mediating Effect of Emotional Intelligence between Physical Exercise and Subjective Well-Being

The research results showed that the emotional intelligence also plays a part in the mediating role in physical exercise and subjective well-being pathways, with a mediation effect ratio of 24.06%. Many studies have shown that physical exercise can bring participants emotional regulation benefits, and the unity and mutual assistance, interpersonal communication, and emotional expression provided by the sports environment were all conducive to the development of emotional intelligence [34,62]. College students with high physical activity have relatively higher emotional intelligence [29], and higher positive emotions. In contrast, the greater the amount of activity, the lower their negative emotion scores [63]. Meanwhile, the improvement of emotional intelligence seems to promote the improvement of subjective well-being. Some scholars pointed out that emotional intelligence plays a positive role in predicting the subjective well-being of an individual, which was manifested as a significant correlation with the cognitive component of subjective well-being, namely life satisfaction, and people with high emotional intelligence who accept, use, understand, and manage their own and others’ emotions were more dominant, and their life satisfaction and subjective well-being levels are also higher [22,25,64,65]. This result was also in line with the opinions of researchers such as Extremera et al. [66], that is, that college students with higher emotional intelligence levels have higher positive emotions, while their negative emotions were lower. Similarly, this study found that in the emotional intelligence of college students, only the emotional management of others and the application of emotion played a part of the mediating role between physical exercise and subjective well-being and its various dimensions, and the level of significance of the emotional management of others was higher. However, there were certain differences in the significance level of each dimension, we speculate that this may be related to the individual differences in the emotional intelligence of college students. Some scholars have pointed out that physical exercise can improve the participants’ enjoyment of the sports experience, provide social support, promote physical changes, enhance physical, mental pleasure and emotional intelligence, thereby improving their quality of life and subjective well-being [52,67].

### 4.4. Chain Mediating Effects of Self-Efficacy and Emotional Intelligence

The research results show that self-efficacy and emotional intelligence play a chain mediating role in the path of physical exercise affecting college students’ subjective well-being, and the chain mediating effect ratio was 8.13%. It was worth noting that, in this process, only the emotional management and emotional application of others have participated in the adjustment of emotional intelligence, while life satisfaction, positive emotions, and negative emotions have all participated in the adjustment of subjective well-being. This shows that college students who regularly participate in physical exercises often have a higher sense of self-efficacy, could be more determined that they could complete a certain behavior and achieve expected goals, were more able to perceive and evaluate the emotions of others in specific situations, could also be correctly applying and managing emotional intelligence, and could better obtain subjective well-being experiences such as positive emotions and life satisfaction. The above and previous studies have confirmed that physical exercise could directly and positively affect the subjective well-being of college students [55,60,61], while self-efficacy plays a mediating role between physical exercise and subjective well-being [59,61], and emotional intelligence plays a mediating role between physical exercise and subjective well-being [52,67], then, if there was a positive correlation between self-efficacy and emotional intelligence, a chain mediating effect may be formed. Interestingly, some studies have shown that self-efficacy and emotional intelligence have a significant correlation, the higher the self-efficacy, the more positive the improvement of individual emotional intelligence [35,36,68,69].

To sum up, physical exercise can not only positively and directly affect the subjective well-being of college students, but also indirectly affect their subjective well-being through the chain mediating effect of self-efficacy and emotional intelligence. Therefore, it is suggested that university administrators should provide students with more physical exercise resources and opportunities, create a good exercise environment, and encourage and guide students to participate in various physical activities, which will help improve their well-being experience and physical and mental health.

### 4.5. Limitations

(1)Since this study was horizontal, the results obtained were more subjective and unable to draw deeper causal relationships. In future studies, longitudinal empirical research could be added to better reveal the causal relationship between variables;(2)This study took students from two comprehensive universities in southwest China as the survey subjects, and the conclusions reached have certain limitations, future studies could select a wider group of subjects to test the external validity of the research results;(3)This study mainly examines the mediating role of self-efficacy and emotional intelligence. In the future, more psychological variables could be added to investigate their impact on the subjective well-being of college students, and the depth and breadth of research could be expanded.

## 5. Conclusions

(1)Physical exercise was significantly positively correlated with college students’ self-efficacy, emotional intelligence, and subjective well-being; self-efficacy was significantly positively correlated with emotional intelligence and subjective well-being; emotional intelligence was significantly positively correlated with subjective well-being;(2)Physical exercise has a direct and positive predictive effect on subjective well-being;(3)Self-efficacy plays a partial mediating role between physical exercise and subjective well-being, and emotional intelligence also plays a partial mediating role between physical exercise and the subjective well-being of college students and its three dimensions; “Self-efficacy → emotional intelligence” has a chain mediating role between physical exercise and subjective well-being. Among them, only others’ emotional management and emotional application were involved in the regulation of emotional intelligence, while life satisfaction, positive emotions, and negative emotions were all involved in the regulation of subjective well-being.

## Figures and Tables

**Figure 1 ijerph-19-11596-f001:**
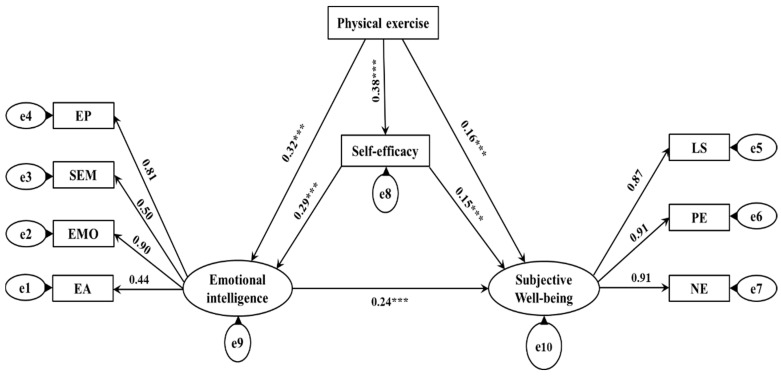
The chain mediation model of self-efficacy and emotional intelligence between physical exercise and subjective well-being. *** *p* < 0.001.

**Table 1 ijerph-19-11596-t001:** Factor extraction and reliability analysis of three measurement scales.

Variables	KMO and Bartlett Ball Inspection	Dimension	Items	Characteristic Root	Explained Variation%	Progressive Explained Variation%	Cronbach’s α
GSES	KMO = 0.893 (*p* < 0.001)	SE	9	4.166	54.158	54.158	0.83
EIS	KMO = 0.875 (*p* < 0.001)	EP	11	7.851	28.365	28.365	0.88
		SEM	6	3.794	10.751	39.116	0.84
		EMO	10	2.513	7.366	46.482	0.91
		EA	6	1.121	4.251	50.733	0.82
SWS	KMO = 0.863 (*p* < 0.001)	LS	5	3.681	52.156	52.156	0.85
	KMO = 0.904 (*p* < 0.001)	PE	9	3.926	38.159	38.159	0.82
		NE	9	1.860	12.451	50.610	0.84

Note: “SE” represents “Self-efficacy”; “EP” represents “Emotional perception”; “SEM” represents “Self-emotion management”; “EMO” represents “Emotion management of others”; “EA” represents “Emotional application”; “LS” represents “Life satisfaction”; “PE” represents “ Positive emotion”; “NE” represents “Negative emotions”.

**Table 2 ijerph-19-11596-t002:** Correlation analysis table of the amount of physical exercise, self-efficacy, emotional intelligence, and subjective well-being of college students (*n* = 826).

Variables	M ± SD	Physical Exercise	Self-Efficacy	Emotional Intelligence	EP	SE	EMO	EA	Subjective Well-Being	LS	NE	PE
Physical exercise	27.24 ± 17.72	1										
Self-efficacy	27.18 ± 11.54	0.38 ***	1									
Emotional intelligence	63.65 ± 13.81	0.34 ***	0.32 ***	1								
EP	15.60 ± 4.66	0.36 ***	0.34 ***	0.80 ***	1							
SE	16.65 ± 5.36	0.19 ***	0.19 ***	0.79 ***	0.38 ***	1						
EMO	13.16 ± 3.59	0.40 ***	0.39 ***	0.78 ***	0.73 ***	0.45 ***	1					
EA	18.27 ± 4.50	0.12 **	0.10 **	0.68 ***	0.38 ***	0.50 ***	0.31 ***	1				
Subjective Well-being	0.00 ± 2.80	0.31 ***	0.30 ***	0.31 ***	0.27 ***	0.20 ***	0.31 ***	0.21 ***	1			
LS	17.45 ± 9.13	0.28 ***	0.26 ***	0.27 ***	0.24 ***	0.17 ***	0.28 ***	0.17 ***	0.92 ***	1		
PE	25.25 ± 11.71	0.29 ***	0.29 ***	0.29 ***	0.25 ***	0.18 ***	0.29 ***	0.19 ***	0.94 ***	0.79 ***	1	
NE	25.58 ± 11.00	0.29 ***	0.29 ***	0.32 ***	0.27 ***	0.21 ***	0.30 ***	0.23 ***	0.94 ***	0.79 ***	0.83 ***	1

Note: ** *p* < 0.01, *** *p* < 0.001.

**Table 3 ijerph-19-11596-t003:** Decomposition of the effect of physical exercise on subjective well-being.

Influence Path	Normalized Effect Size	Ratio of Total Effect	Significance
Direct effect	0.16	50.00%	Significant
Physical exercise → self-efficacy → subjective well-being	0.38 × 0.15 = 0.057	17.81%	Significant
Physical exercise → emotional intelligence → subjective well-being	0.32 × 0.24 = 0.077	24.06%	Significant
Physical exercise → self-efficacy → emotional intelligence → subjective well-being	0.38 × 0.29 × 0.24 = 0.026	8.13%	Significant
Total indirect effect	0.057 + 0.077 + 0.026 = 0.16	50.00%	Significant
Total effect	0.16 + 0.16 = 0.32	—	Significant

**Table 4 ijerph-19-11596-t004:** Regression analysis between the sub-dimensions of physical exercise, self-efficacy, and subjective well-being.

Variables	Dimension	Physical Exercise	Self-Efficacy	R	R^2^	F
Subjective well-being	LS	0.279 ***	0.180 ***	0.279	0.078	69.300 ***
	PE	0.286 ***		0.286	0.082	73.646 ***
	NE	0.294 ***		0.294	0.086	78.004 ***
Self-efficacy		0.382 ***		0.382	0.146	140.411 ***
Subjective well-being	LS	0.210 ***		0.324	0.105	48.413 ***
	PE	0.208 ***	0.205 ***	0.344	0.118	55.097 ***
	NE	0.216 ***	0.204 ***	0.349	0.122	57.138 ***

Note: *** *p* < 0.001.

**Table 5 ijerph-19-11596-t005:** Regression analysis among sub-dimensions of physical exercise, emotional intelligence, and subjective well-being.

Variables	Dimension	Physical Exercise	Emotional Intelligence				R	R^2^	F
			EP	SEM	EMO	EA			
Subjective well-being	LS	0.279 ***					0.279	0.078	69.300 ***
	PE	0.286 ***					0.286	0.082	73.646 ***
	NE	0.294 ***					0.294	0.086	78.004 ***
Emotional intelligence	EP	0.356 ***					0.356	0.127	119.611 ***
	SEM	0.185 ***					0.185	0.034	29.163 ***
	EMO	0.398 ***					0.398	0.159 ***	155.496 ***
	EA	0.115 **					0.115	0.013	11.047 **
Subjective well-being	LS	0.198 ***	0.007	0.017	0.165 **	0.084 *	0.347	0.120	22.447 ***
	PE	0.201 ***	0.012	0.009	0.168 **	0.105 **	0.363	0.132	24.916 ***
	NE	0.205 ***	0.036	0.027	0.141 **	0.130 **	0.384	0.147	28.335 ***

Note: * *p* < 0.05, ** *p* < 0.01, *** *p* < 0.001, m1 represents emotional perception, m2 represents self-emotion management, m3 represents emotional management of others, and m4 represents the emotional application.

**Table 6 ijerph-19-11596-t006:** Regression analysis among sub-dimensions of physical exercise, self-efficacy, emotional intelligence, and subjective well-being.

Variables	Dimension	Physical Exercise	Self-Efficacy	Emotional Intelligence				R	R^2^	F
				EP	SEM	EMO	EA			
Subjective well-being	LS	0.162 ***	0.136 ***	−0.006	0.012	0.136 **	0.090 *	0.367	0.135	21.306 ***
	PE	0.159 ***	0.161 ***	−0.003	0.003	0.135 **	0.113 **	0.390	0.152	24.529 ***
	NE	0.164 ***	0.157 ***	0.021	0.022	0.108 *	0.137 ***	0.408	0.167	27.298 ***

Note: * *p* < 0.05, ** *p* < 0.01, *** *p* < 0.001.

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
