# Peer review of "The Relationship among College Students’ Physical Exercise, Self-Efficacy, Emotional Intelligence, and Subjective Well-Being"

_ijerph, 2022, doi:10.3390/ijerph191811596_

Round 1

Reviewer 1 Report

This study heavily relies on statistical analysis to look for relationships between four variables. While the whole design, analysis and findings are valid, there are rooms to improve. Two major weaknesses are: 1) your measured self-efficacy is not physical activity (PA) specific which makes this study a regular one rather than an in-depth one. So overall well-being, emotional intelligence, and self-efficacy are all broadly defined; and (2) when all these are broad social/psychological constructs, their predictive values to physical activities may not be able to compete with those more PA specific factors. For example, availability of PA space, time, and resource, peer influence or social support, to name a few. It makes this study less significant.  

In term of the manuscript, it has a clear description about what to study about, the four variables and their relationships, especially about the mediating effect or more precisely speaking, the indirect effect. My comments are sorted by manuscript sections.

Introduction Section

My first comment goes to the word of choice in the introduction section. It is a common practice in journal articles though. So my comment is more a call for attention. 

The comment is about your last paragraph of introduction. I picked up a few wordings that I like to suggest for a revision: 

"The interaction between different factors has not been fully demonstrated and elucidated."

You have cited articles that studied the matter before but you think they did not do a fully demonstration. What is a fully demonstration? Maybe previous studies did not use a sophisticated model in analysis. You may just point that out so readers will understand what is a fully demonstration and what is not. 

"few scholars have conducted in-depth research on the overall path of “physical exercise, self-efficacy, emotional intelligence, and subjective well-being” of college students. 

in-depth and overall path are two subjective descriptions too. What do they mean? A path analysis to study saturated relationships?

"aims to reveal the deep correlations between various variables, deepen the inherent benefits of exercise psychology"

What does deep correlation mean? Is there a correlation that can be called shallow? I don't see how you can deepen the correlation with the four variables under study. You may be able to explore different ways to study the correlation. 

Methodology section

Did you get IRB approved? Was the study funded? 

Participants heading

I do not see any description as how samples are randomly selected.

No data collection procedures are introduced. 

Questionnaire design subheading

You have described each scale that has been used in detail. I suggest adding a short description at the beginning of each scale section to indicate what the scale is used for. 

For example,  the Physical Activity Rating Scale (PARS-3) is the one designed for categorizing a person's level of physical activity. I think it is quite common that  Physical Activity is defined as the level of activity.

As for your emotional intelligence scale and self-efficacy scale, you may also include a simple introduction to define what is the emotional intelligence you are studying. Both emotional intelligence and self-efficacy are broad concepts. How you measure them suggests what you are studying. There are many scales out there for selection. Different scales are constructed based on different definitions or domains/subdomains. Giving an introduction may provide information as to what you are targeting specifically. For example, you described self-efficacy (in your introduction) as “Self-efficacy refers to the subjective evaluation of an individual’s ability or grasp of his ability to complete a specific behavior in a specific situation.” While your measurement of self-efficacy is not the one designed for a specific behavior. So it is not physical activity specific.

You used the term “Measurement model verification”. Please indicate what the Measurement model is.  

Data analysis subheading

Should give more information. In your case, since you have three hypotheses to test, you can explain which one is tested by which method and why. I am the one who does not believe more statistical methods applied will make a manuscript better. So, you may want to explain your major stats applications. You have valid scales. Why do you need to do factor analysis and use it to define subcategories, for instance? Didn't the author have specify dimensions of the scale and how to make composite scores of the scale? 

Result

I look through the result and discussion sections. There are wordings that I disagree on. Overall report is good. I did not match your result and discussion to your proposed three hypotheses. But key things around the four variables, and intended test of their relationships have been focused on, which is good. 

3.2. The Mediating Effect of 

“structural equation model”. Strictly speaking you did not use a structural equation model, you used a path analysis.

Author Response

Reviewer 1:

This study heavily relies on statistical analysis to look for relationships between four variables. While the whole design, analysis and findings are valid, there are rooms to improve. Two major weaknesses are: 1) your measured self-efficacy is not physical activity (PA) specific which makes this study a regular one rather than an in-depth one. So overall well-being, emotional intelligence, and self-efficacy are all broadly defined; and (2) when all these are broad social/psychological constructs, their predictive values to physical activities may not be able to compete with those more PA specific factors. For example, availability of PA space, time, and resource, peer influence or social support, to name a few. It makes this study less significant.

In term of the manuscript, it has a clear description about what to study about, the four variables and their relationships, especially about the mediating effect or more precisely speaking, the indirect effect. My comments are sorted by manuscript sections.

Introduction Section

The comment is about your last paragraph of introduction. I picked up a few wordings that I like to suggest for a revision: "The interaction between different factors has not been fully demonstrated and elucidated." You have cited articles that studied the matter before but you think they did not do a fully demonstration. What is a fully demonstration? Maybe previous studies did not use a sophisticated model in analysis. You may just point that out so readers will understand what is a fully demonstration and what is not.

Reply: Thank you very much for your suggestion, we have corrected it accordingly in the manuscript.

"few scholars have conducted in-depth research on the overall path of “physical exercise, self-efficacy, emotional intelligence, and subjective well-being” of college students. in-depth and overall path are two subjective descriptions too. What do they mean? A path analysis to study saturated relationships?

Reply: Thank you very much for your suggestion, we have corrected it accordingly in the manuscript.

"aims to reveal the deep correlations between various variables, deepen the inherent benefits of exercise psychology". What does deep correlation mean? Is there a correlation that can be called shallow? I don't see how you can deepen the correlation with the four variables under study. You may be able to explore different ways to study the correlation.

Reply: Thank you very much for your suggestion, there is some misrepresentation here and we have corrected it.

Methodology section

Did you get IRB approved? Was the study funded?

Reply: The study was approved accordingly, and funding information has been placed after the main text and before references.

Participants heading

I do not see any description as how samples are randomly selected. No data collection procedures are introduced.

Reply: We have supplemented the random sampling process and data collection procedure in the manuscript, please consult the manuscript.

Questionnaire design subheading

You have described each scale that has been used in detail. I suggest adding a short description at the beginning of each scale section to indicate what the scale is used for. For example, the Physical Activity Rating Scale (PARS-3) is the one designed for categorizing a person's level of physical activity. I think it is quite common that Physical Activity is defined as the level of activity.

As for your emotional intelligence scale and self-efficacy scale, you may also include a simple introduction to define what is the emotional intelligence you are studying. Both emotional intelligence and self-efficacy are broad concepts. How you measure them suggests what you are studying. There are many scales out there for selection. Different scales are constructed based on different definitions or domains/subdomains. Giving an introduction may provide information as to what you are targeting specifically. For example, you described self-efficacy (in your introduction) as “Self-efficacy refers to the subjective evaluation of an individual’s ability or grasp of his ability to complete a specific behavior in a specific situation.” While your measurement of self-efficacy is not the one designed for a specific behavior. So it is not physical activity specific.

Reply: Thank you very much for your suggestion, we have revised and improved the questionnaire part.

You used the term “Measurement model verification”. Please indicate what the Measurement model is.

Reply: “Measurement model verification” refers “The confirmatory factor analysis”, we have corrected the representation。

Data analysis subheading)

Should give more information. In your case, since you have three hypotheses to test, you can explain which one is tested by which method and why. I am the one who does not believe more statistical methods applied will make a manuscript better. So, you may want to explain your major stats applications. You have valid scales. Why do you need to do factor analysis and use it to define subcategories, for instance? Didn't the author have specify dimensions of the scale and how to make composite scores of the scale?

Reply: Thank you very much for your suggestion, we have reworked the data analysis section. It should be pointed out that although we use a mature scale, many studies believe that there may be differences in the reliability, validity and factor dimensions of the scale for different populations. Therefore, it is recommended to use factor analysis and Cronbach α to test the scale.

Result

I look through the result and discussion sections. There are wordings that I disagree on. Overall report is good. I did not match your result and discussion to your proposed three hypotheses. But key things around the four variables, and intended test of their relationships have been focused on, which is good.

Reply: Thanks for your recognition.

3.2. The Mediating Effect of “structural equation model”. Strictly speaking you did not use a structural equation model, you used a path analysis.

Reply: Thanks for the suggestion, but it's important to point out that our model includes both observable explicit variables and latent variables that cannot be directly observed. So, as far as we know, this is characteristic of structural equation modeling.

Reviewer 2 Report

Thank you for the Authors contribution to this study. I have the following suggestions for the authors.

  • The manuscript is well introduced, but I recommend to the authors to use a hypothesized path model since it would be more sufficient to highlight the main direction and hypothesis of this study.   
  • It is not clear why the authors used factor analysis on the previously validated scales. I believe reporting Cronbach alphas or other reliability values would be enough in this case.
  • Include table 1 to the end of subchapter 1.2.1
  • Data analysis is missing the methods for the measurement scales (EFA, CFA)
  • On table 2 "Subjective Well-being" has a mean of 0.00. Is this correct?
  • Please reconstruct the result of the SEM (figure 1). Using the figure from AMOS is not professional.  
  • Regression analysis was also used in this study. Please include it in the data analysis and explain why you used regression analysis as well. Regression analysis in this study seems unnecessary since SEM was used to identify the relationship between these results.
  • The discussion and the conclusion are well written. I believe no changes need in this part.

Author Response

Reviewer 2:

Thank you for the Authors contribution to this study. I have the following suggestions for the authors.

The manuscript is well introduced, but I recommend to the authors to use a hypothesized path model since it would be more sufficient to highlight the main direction and hypothesis of this study.

Reply: Thanks for the suggestion, but it's important to point out that our model includes both observable explicit variables and latent variables that cannot be directly observed. So, as far as we know, this is characteristic of structural equation modeling.

It is not clear why the authors used factor analysis on the previously validated scales. I believe reporting Cronbach alphas or other reliability values would be enough in this case.

Reply: Although we use a priori scale, previous studies have pointed out that the reliability and validity of the scale are easily affected by factors such as population and region. Therefore, to ensure the applicability of the scale, factor analysis of the scale should be carried out again to determine the factor dimensions in the research, and the reliability and validity of the scale should be examined in combination with the Cronbach alphas coefficient.

Include table 1 to the end of subchapter 1.2.1

Reply: Thanks for your suggestion, we have corrected it in the manuscript.

Data analysis is missing the methods for the measurement scales (EFA, CFA)

Reply: Thanks for your suggestion, we have corrected it in the manuscript.

On table 2 "Subjective Well-being" has a mean of 0.00. Is this correct?

Reply: According to previous research, the total score of subjective well-being is the sum of Z scores after reverse scoring of life satisfaction, positive affect and negative affect, and the range of scores involves positive and negative numbers, so the mean value involves 0.

Please reconstruct the result of the SEM (figure 1). Using the figure from AMOS is not professional.

Reply: Thanks for your suggestion, we have reconstruct the result of the SEM in the manuscript.

Regression analysis was also used in this study. Please include it in the data analysis and explain why you used regression analysis as well. Regression analysis in this study seems unnecessary since SEM was used to identify the relationship between these results.

Reply: Thanks for your suggestion, we have included regression analysis in our data analysis. At the same time, the purpose of using regression analysis is to more intuitively reflect the relationship between the main variables and each sub-dimension of subjective well-being, and to explain the model in more detail.

The discussion and the conclusion are well written. I believe no changes need in this part.

Reply: Thank you for your recognition.

Reviewer 3 Report

The overall study is sound and coherent from the methodology and interpretation standpoint. The results are perfectly coherent and the discussion further explores this important topic. Overall, I think the manuscript only needs some minor adjustments before publication:

  • in the General Self-Report Scale (Method section), the option 'very agree' is incorrect. Please reformulate
  • in the Research Results section, point 2.3, the first sentence seems incomplete. This happens some times across the manuscript;
  • Figure 1 and Table 6 need a note on the significance levels;
  • this section also need reference to figures and tables across the text, following each description
  • in the Discussion section, 1st paragraph, the authors say college students who participate in sports activities score higher in (...) negative emotions. We should read they score lower in negative emotions. Please reformulate

There are some incoherencies in the phrases across the manuscript, although the overall content is perceivable. The document should be proofread by a native English to ensure correct readability.

Author Response

Reviewer 3:

The overall study is sound and coherent from the methodology and interpretation standpoint. The results are perfectly coherent and the discussion further explores this important topic. Overall, I think the manuscript only needs some minor adjustments before publication:

Reply: Thank you for your recognition.

in the General Self-Report Scale (Method section), the option 'very agree' is incorrect. Please reformulate in the Research Results section, point 2.3, the first sentence seems incomplete. This happens some times across the manuscript;

Reply: Thanks for your suggestion, we have corrected the option 'very agree'. Also rephrased the first sentence of Section 2.3 in the manuscript.

Figure 1 and Table 6 need a note on the significance levels; this section also need reference to figures and tables across the text, following each description in the Discussion section, 1st paragraph, the authors say college students who participate in sports activities score higher in (...) negative emotions. We should read they score lower in negative emotions. Please reformulate

Reply: Thanks for your suggestion, we have supplemented the significance levels in the corresponding sections in the manuscript. Meanwhile, we have also corrected language errors in the manuscript.

There are some incoherencies in the phrases across the manuscript, although the overall content is perceivable. The document should be proofread by a native English to ensure correct readability.

Reply: Thanks for your suggestion, we have invited English majors to revise the manuscript, and I believe there will be some improvement.

Reviewer 4 Report

Thank you for the opportunity to review this paper. First, I would like to appreciate the authors' effort to address the role of sport participation in affecting college students' emotions and well-being. College students have been a critical population in the field of sport & health studies. The findings of this study highlight the importance of sport participation to their psychological outcomes. However, I would like to ask the authors to address some of my comments and suggestions. 

First, as I mentioned before, college students have been an important segment in sport research. However, the authors did not address the uniqueness and importance of college students. That is, why physical exercise, emotional intelligence, and well-being are important and relevant to college students? This needs to be addressed in the introduction. 

Second, because the research model has been developed, the "Research model and hypothesis development" section is necessary. Moreover, the literature review is missing. The authors need to restructure the introduction and add the "Literature review" and "Research model and hypothesis development" sections before methods. 

Moreover, I found that most references are studies in Chinese. Please cite more international studies. I found some studies which can help the development of research model and hypotheses. Please consider and find references below. 

Shang, Y., Xie, H. D., & Yang, S. Y. (2021). The relationship between physical exercise and subjective well-being in college students: The mediating effect of body image and self-esteem. Frontiers in Psychology12, 1776.

Graupensperger, S., Panza, M. J., Budziszewski, R., & Evans, M. B. (2020). Growing into “Us”: Trajectories of Social Identification with College Sport Teams Predict Subjective Well‐Being. Applied Psychology: Health and Well‐Being12(3), 787-807.

Miller, K. E., & Hoffman, J. H. (2009). Mental well-being and sport-related identities in college students. Sociology of sport journal26(2), 335.

Shin, S., Chiu, W., & Lee, H. W. (2018). For a better campus sporting experience: Scale development and validation of the collegiate sportscape scale. Journal of Hospitality, Leisure, Sport & Tourism Education22, 22-30.

Shin, S., Chiu, W., & Lee, H. W. (2019). Impact of the social benefits of intramural sports on Korean students’ quality of college life and loyalty: A comparison between lowerclassmen and upperclassmen. The Asia-Pacific Education Researcher28(3), 181-192.

There are more references you should include in your research. 

The results section is satisfactory and clearly interpreted. 

The discussion section should have stronger theoretical implications.  Moreover, practical implications should be made for university administrators.

Author Response

Reviewer 4:

Thank you for the opportunity to review this paper. First, I would like to appreciate the authors' effort to address the role of sport participation in affecting college students' emotions and well-being. College students have been a critical population in the field of sport & health studies. The findings of this study highlight the importance of sport participation to their psychological outcomes. However, I would like to ask the authors to address some of my comments and suggestions.

First, as I mentioned before, college students have been an important segment in sport research. However, the authors did not address the uniqueness and importance of college students. That is, why physical exercise, emotional intelligence, and well-being are important and relevant to college students? This needs to be addressed in the introduction.

Reply: Thanks for your suggestion, we have supplemented the corresponding content in the Introduction.

Second, because the research model has been developed, the "Research model and hypothesis development" section is necessary. Moreover, the literature review is missing. The authors need to restructure the introduction and add the "Literature review" and "Research model and hypothesis development" sections before methods. (其次, Reply: Thanks for your suggestion, we have made appropriate corrections to the manuscript, but do not know whether it conforms to your opinion. If there is any inappropriateness, please point out, thanks.

Moreover, I found that most references are studies in Chinese. Please cite more international studies. I found some studies which can help the development of research model and hypotheses. Please consider and find references below.

Shang, Y., Xie, H. D., & Yang, S. Y. (2021). The relationship between physical exercise and subjective well-being in college students: The mediating effect of body image and self-esteem. Frontiers in Psychology, 12, 1776.

Graupensperger, S., Panza, M. J., Budziszewski, R., & Evans, M. B. (2020). Growing into “Us”: Trajectories of Social Identification with College Sport Teams Predict Subjective Well‐Being. Applied Psychology: Health and Well‐Being, 12(3), 787-807.

Miller, K. E., & Hoffman, J. H. (2009). Mental well-being and sport-related identities in college students. Sociology of sport journal, 26(2), 335.

Shin, S., Chiu, W., & Lee, H. W. (2018). For a better campus sporting experience: Scale development and validation of the collegiate sportscape scale. Journal of Hospitality, Leisure, Sport & Tourism Education, 22, 22-30.

Shin, S., Chiu, W., & Lee, H. W. (2019). Impact of the social benefits of intramural sports on Korean students’ quality of college life and loyalty: A comparison between lowerclassmen and upperclassmen. The Asia-Pacific Education Researcher, 28(3), 181-192.

There are more references you should include in your research.

Reply: Thanks for your suggestion, we have appropriately cited parts of the literature in the manuscript.

The results section is satisfactory and clearly interpreted.

Reply: Thank you for your recognition.

The discussion section should have stronger theoretical implications.  Moreover, practical implications should be made for university administrators.

Reply: Thank you for your recognition. We have enriched and refined the Discussion section.

Round 2

Reviewer 1 Report

I have a few comments in method section. Those are minor revisions needed, to my point of view. 

Method

"The students in grades 1 to 4 from two comprehensive universities". Maybe change 1 to 4 to freshmen to seniors which will be clearer to readers of international space. 

How the number of 110 to 130 per grade is determined? Was 100 almost 10% of the total students in a grade? You can report your population size which would be the number of students in the two universities from freshman to senior grades. How did you randomly select your samples and then invite them to participate? Did you conduct the survey in the classroom or send questionnaires to the students to take home? It sounds like you did an in-person (vs. online) self-administered questionnaire survey. You have two universities with eight grades in total. it seems reasonable you end up with 1005 surveys if roughly 100 per grade. Just need some information.   

I am not clear what variables are used in analysis. The variables you write in your syntax. I use this following example to make my point. You wrote:

"Physical Activity Rating Scale (PARS-3) The Physical Activity Rating Scale (PARS-3) is the one designed for categorizing a person’s level of physical activity. The PARS-3 was revised by Liang [35], namely physical exercise intensity, exercise time, and exercise frequency. Using Likert’s 5-point formula for quantification, with a score ranging from 1 to 5 points. Physical exercise score = exer-cise intensity score × (exercise time score-1) × exercise frequency score, the score interval 0-100 points, according to which to measure the level of participation in physical exercise. The evaluation criteria of exercise volume were: low exercise volume ≤19 points, medium exercise volume 20-42 points, and high exercise volume ≥43 points. The test-retest relia-bility of the scale was relatively high, and the correlation coefficient r=0.82. "

So by your description I know the instrument, number of questions, the scale, and your test result for reliability, etc. But what is the final variable you produced and used in your model test(s)?  It could be the physical exercise score as you mentioned. If that is true, what is the score range in theory and what is the score range you obtained from your samples? It is a continuous variable for sure. You also mentioned the criteria of exercise volume. So did you actually use a categorical variable in analysis? 

I actually have to guess what are the final variables included in your analysis, those composite scores, for all the scales or instruments you studied. It will be nice to introduce the final variable, its baseline parameters such as means and standard deviations.

It will also be nice to introduce briefly who the samples are: females, males, and grade distributions.   

Author Response

Reviewer 1

"The students in grades 1 to 4 from two comprehensive universities". Maybe change 1 to 4 to freshmen to seniors which will be clearer to readers of international space.

Reply: Thanks for your suggestion, we have revised this content.

How the number of 110 to 130 per grade is determined? Was 100 almost 10% of the total students in a grade? You can report your population size which would be the number of students in the two universities from freshman to senior grades. How did you randomly select your samples and then invite them to participate? Did you conduct the survey in the classroom or send questionnaires to the students to take home? It sounds like you did an in-person (vs. online) self-administered questionnaire survey. You have two universities with eight grades in total. it seems reasonable you end up with 1005 surveys if roughly 100 per grade. Just need some information.

Reply: Thanks for your suggestion, we have added corresponding instructions in the Methods section.

I am not clear what variables are used in analysis. The variables you write in your syntax. I use this following example to make my point. You wrote:

"Physical Activity Rating Scale (PARS-3) The Physical Activity Rating Scale (PARS-3) is the one designed for categorizing a person’s level of physical activity. The PARS-3 was revised by Liang [35], namely physical exercise intensity, exercise time, and exercise frequency. Using Likert’s 5-point formula for quantification, with a score ranging from 1 to 5 points. Physical exercise score = exer-cise intensity score × (exercise time score-1) × exercise frequency score, the score interval 0-100 points, according to which to measure the level of participation in physical exercise. The evaluation criteria of exercise volume were: low exercise volume ≤19 points, medium exercise volume 20-42 points, and high exercise volume ≥43 points. The test-retest relia-bility of the scale was relatively high, and the correlation coefficient r=0.82. "

So by your description I know the instrument, number of questions, the scale, and your test result for reliability, etc. But what is the final variable you produced and used in your model test(s)?  It could be the physical exercise score as you mentioned. If that is true, what is the score range in theory and what is the score range you obtained from your samples? It is a continuous variable for sure. You also mentioned the criteria of exercise volume. So did you actually use a categorical variable in analysis?

Reply: The final variable generated and used in model testing is physical activity score, and we have supplemented the manuscript with theoretical and actual score ranges. We did not use classified physical activity scores in our study, so we chose to delete the classification.

I actually have to guess what are the final variables included in your analysis, those composite scores, for all the scales or instruments you studied. It will be nice to introduce the final variable, its baseline parameters such as means and standard deviations. It will also be nice to introduce briefly who the samples are: females, males, and grade distributions.

Reply: Thanks for your suggestion, we've made some additions appropriately. Among them, the distribution of gender and grade has been explained in section 1.1 1.1 Participants, and the parameters introduced into the final variable are shown in section 2.2 with the mean and standard deviation of each variable.

Reviewer 2 Report

Thank you for the Authors contribution! All changes have been made! I recommend publishing this paper.

Author Response

Thank you for the Authors contribution! All changes have been made! I recommend publishing this paper.

Reply:Thank you very much for your review and endorsement!

Reviewer 4 Report

I would like to appreciate the authors' efforts to revise the manuscript. I found the authors satisfactorily addressed all reviewers' comments. Good job. 

Author Response

I would like to appreciate the authors' efforts to revise the manuscript. I found the authors satisfactorily addressed all reviewers' comments. Good job. 

Reply:Thank you very much for your review and endorsement!